# Ultrasound Features of *Helicobacter pylori*-Related Gastritis

**DOI:** 10.3390/antibiotics14010012

**Published:** 2024-12-28

**Authors:** Fulvia Terracciano, Antonella Marra, Veronica Nassisi, Chiara Lillo, Fabrizio Bossa, Sonia Carparelli, Francesco Cocomazzi, Maria Rosa Valvano, Giuseppe Losurdo, Alfredo Di Leo, Francesco Perri

**Affiliations:** 1Division of Gastroenterology & Endoscopy, IRCCS, Fondazione “Casa Sollievo della Sofferenza”, 71013 San Giovanni Rotondo, Italy; marra.antonella_1988@libero.it (A.M.); chiaralillo13@gmail.com (C.L.); f.bossa@operapadrepio.it (F.B.); carparelli.sonia@gmail.com (S.C.); francescococomazzi@gmail.com (F.C.); f.perri@operapadrepio.it (F.P.); 2National Institute of Gastroenterology, IRCCS Saverio de Bellis Research Hospital, 70013 Castellana Grotte, Italy; 3Internal Medicine Unit, Department of Clinical and Experimental Medicine, University of Messina, 98122 Messina, Italy; veronica.nassisi@gmail.com; 4Quality and Accreditation Unit, IRCCS, Fondazione “Casa Sollievo della Sofferenza”, 71013 San Giovanni Rotondo, Italy; mr.valvano@operapadrepio.it; 5Section of Gastroenterology, Department of Precision and Regenerative Medicine and Ionian Area, University of Bari, 70124 Bari, Italyalfredo.dileo@uniba.it (A.D.L.)

**Keywords:** *H. pylori* gastritis, abdominal ultrasound, submucosal wall thickness, gastric motility

## Abstract

**Background** Abdominal ultrasound (US) is a reliable method for visualizing gastric wall layers and measuring their thickness. The objective of this study is to characterize the ultrasound features of *H. pylori*-induced gastritis and assess its predictive potential role for this condition. **Methods** A cohort of 119 patients underwent gastroscopy with biopsy and abdominal US to evaluate antral wall thickness (AWT), submucosal wall thickness (SLT), mucosal wall thickness (MLT), gastric motility, and the presence of ingested material. They were divided into three groups: group A (normal mucosa without *H. pylori* infection), group B (gastritis *H. pylori* negative) and group C (gastritis *H. pylori* positive). **Results**: SLT and reduced gastric motility were significantly increased in the *H. pylori* gastritis group (*p* < 0.001). Multivariate analysis identified SLT as the only significant independent predictor of *H. pylori* gastritis (*p* < 0.001). An SLT threshold of 1.55 mm was determined as optimal for differentiating *H. pylori*-positive patients from -negative, yielding a sensitivity of 77% and a specificity of 72%. **Conclusions** These findings suggest that *H. pylori* gastritis is characterized by submucosal layer thickening and impaired gastric motility.

## 1. Introduction

*Helicobacter pylori* (*H. pylori*) is the most prevalent chronic bacterial infection transmitted from human to human. *H. pylori* is a cause of gastritis and bleeding ulcers, but various studies have investigated its possible role in dyspepsia and oral cavity disorders such as periodontitis [1,2].

The diagnostic evaluation for *H. pylori* can be performed using either invasive (endoscopic) or noninvasive techniques, depending on the clinical setting. The choice of method is guided by whether the patient requires upper endoscopy for symptom investigation or ongoing monitoring. In cases where endoscopy is not necessary, the preferred approach to diagnosing *H. pylori* infection involves noninvasive tests, such as the stool antigen test or the urea breath test (UBT), both of which offer high accuracy in detecting active infection [3]. These tests are primarily recommended for conditions linked to *H. pylori*, including active peptic ulcers, chronic gastritis, gastric cancer, and MALT lymphoma. They may also be considered, though with less robust evidence, for uninvestigated dyspepsia in individuals under 60 years of age without alarming symptoms, or in cases of prolonged NSAID use, unexplained anemia due to iron deficiency, idiopathic thrombocytopenic purpura (ITP), or vitamin B12 deficiency [3].

*H. pylori* gastritis often begins with a localized infection predominantly affecting the antral region of the stomach, with limited involvement of the gastric body. At this stage, there is an increase in gastrin production and a decrease in somatostatin secretion, leading to elevated acid secretion, which can contribute to ulcer formation in some cases [4].

If the inflammatory process persists, there is a gradual loss of both gastrin-producing cells and acid-secreting parietal cells, resulting in reduced acid secretion and the development of gastric atrophy accompanied by intestinal metaplasia [5]. These pathological changes enable the bacteria to migrate proximally, leading to inflammation in the gastric corpus. In certain cases, the disease progresses to a widespread inflammation involving both the antrum and the corpus, eventually resulting in atrophic gastritis with a marked reduction in acid secretion [6].

To date, the indications of the gastric examination by transabdominal ultrasound have been limited. Nevertheless, the stomach can be studied, starting from the gastroesophageal junction up to the pyloric ring with a convex and linear probe with a frequency between 3.5 and 13 MHz. Ultrasound distinguishes five wall layers with different echogenicity to the non-pathological stomach. From the inside out, the first hyperechoic layer represents the interface between the intraluminal contents and the gastric mucosa, while the second hypoechoic layer corresponds to the mucosa. The third hyperechoic layer is associated with the submucosa, the fourth hypoechoic layer corresponds to the muscularis propria, and the fifth hyperechoic layer represents the serosa. This imaging technique can also be utilized to assess various aspects of gastric function, including antral contractility, gastric emptying, transpyloric flow, gastric accommodation, and the intragastric distribution and volume of meals [7].

The objective of this study is to evaluate the ultrasonographic characteristics of antral gastritis and determine any differences between patients without histological evidence of gastritis (therefore with normal gastric mucosa) and between patients with antral gastritis, in turn with and without *H. pylori* infection.

## 2. Results

The total number of screened patients was 119.

Several histological patterns have been identified: chronic active gastritis, chronic atrophic gastritis, chronic inactive gastritis, active gastritis, PPI-induced damages, gastritis with metaplasia areas, autoimmune gastritis. Six of them had atrophic gastritis on histology, therefore they were excluded according to selection criteria. Hence the total number of patients analyzed is 113, divided into 3 groups as follows:-Group 0: 35 patients without evidence of gastritis on histological examination;-Group 1: 40 patients with positive histology for gastritis, in the absence of *Helicobacter pylori;*-Group 2: 38 patients with positive histology for *H. pylori*-related gastritis.

The three groups were compared for demographic characteristics: sex, age, smoking habit, alcohol consumption, NSAID use, and PPI use (Table 1).

No statistically significant differences were found regarding demographic variables among the three groups.

The three groups were compared for ultrasound characteristics: submucosal thickness, mucosal thickness, the ratio between submucosa and the thickness of the entire gastric wall at the antral level, reduced gastric motility, and the presence of delayed gastric emptying (Table 2).

The submucosal thickness and reduced motility were significantly higher in the group of patients with histological confirmation of *Helicobacter pylori*-related gastritis (*p* < 0.001) (Figure 1 and Figure 2a,b).

Variables that were statistically significant in the univariate analysis (submucosal thickness and reduced motility) were tested in the multivariate analysis using a logistic regression model with backward stepwise term selection, and there was only one. In the multivariate analysis, the reference category was group 0 (patients with negative histology for gastritis), and was compared, respectively, with patients from group 1 and group 2 (Table 3).

In the multivariate analysis, only the increased submucosal thickness emerged as an independent predictive factor for *H. pylori*-related gastritis. Specifically, this characteristic was statistically more significant in patients belonging to group 2 (patients with positive histology for *H. pylori*-related gastritis) with a *p*-value < 0.001.

A receiver operating characteristic (ROC) curve was constructed to establish a minimum threshold value of submucosal thickness to better distinguish between *H. pylori*-positive (group 2) and -negative (group 1) patients (Figure 3).

The optimal threshold value for submucosal thickness was defined as 1.55 mm, with an area under the curve of 0.7544 (*p* = 0.000009), yielding a sensitivity of 77% and a specificity of 72%.

Among the 38 patients in group 2, three patients decided not to take antibiotic therapy. Of the remaining 35 patients, 29 took bismuth quadruple therapy, and 6 patients took concomitant therapy. We did not observe any drop out, therefore the per-protocol and intention-to-treat analysis coincided. All six patients receiving concomitant regimen successfully eradicated the germ. We observed only one failure among the bismuth quadruple therapy group; therefore, the eradication rate was 96.5%. The failure patient subsequently took concomitant triple therapy with success. Regarding side effects, we recorded only two cases of dysgeusia in the Bismuth quadruple group (6.9%). Further details about eradication are summarized in Table 4.

## 3. Discussion

Dyspepsia is a common symptom with a broad differential diagnosis and heterogeneous pathophysiology [8]. The prevalence of functional dyspepsia ranges from 5 to 11% worldwide [9] and can significantly impact on the quality of life [10]. Among the known causes of dyspepsia, *Helicobacter pylori* infection is quite common. The gold standard for diagnosing *H. pylori* infection remains biopsy sampling through esophagogastroduodenoscopy [11]; it allows us to exclude the presence of protruding masses and to establish the presence of ulcers, gastritis, and evaluate the presence or absence of atrophy/metaplasia. However, it is an invasive method, burdened by high costs, patient fear, and potential complications [12]. Other non-invasive methods, such as the UBT and fecal antigen stool detection, are highly sensitive and specific in detecting *Helicobacter pylori* infection; however, they do not allow us to exclude the presence of complications of gastritis, nor (pre)-neoplastic conditions; furthermore, they do not detect the presence or degree of inflammation, nor the grade of atrophy/metaplasia. Furthermore, neither endoscopy nor non-invasive tests allow for studying gastric motility and emptying times.

Ultrasound of the gastrointestinal tract is a real-time, dynamic, non-invasive, repeatable method, with almost no complications for the patient: it allows for study of the stomach in all its portions, from the gastroesophageal junction to the pylorus, throughout its thickness, from the lumen to the serosa, allowing for the identification of both protruding and submucosal masses.

Ultrasound is also useful for analyzing gastric content and thus the presence of ingestions, gastric motility, and emptying time for the diagnosis of gastroparesis. In our study, we aimed to investigate the presence of ultrasound signs predictive of chronic gastritis and, in particular, chronic gastritis due to *H. pylori*. Initially, *H. pylori* localizes at the level of the antral mucosa causing erosions, and in response, the mucosa thickness; subsequently, there is a concomitant thickening of this layer parallel to the extent and severity of the inflammatory changes. The thickening of the submucosal layer in patients with *H. pylori* infection that emerged in our study could be justified by the chronicization of the inflammatory process caused by the bacterium itself.

Few studies have investigated the presence of ultrasound signs of Helicobacter pylori gastritis. In 2010, Mazaher et al. [13] evaluated 100 children aged 1 to 15 years by ultrasound and compared the thicknesses of the various layers of the gastric antrum and duodenum between children with *H. pylori* infection and those without infection: the mean thickness of the mucosa and the sum of the mucosa and submucosa in both the gastric antrum and duodenal bulb were significantly higher in patients with bacterial infection than in those without infection.

Cakmakci et al. in 2013 [14] and Zaher T. et al. in 2020 [15], respectively, divided 108 and 90 patients into 3 groups: a control group with asymptomatic patients and no evidence of *H. pylori*-related gastritis, a group with Helicobacter pylori-negative gastritis, and a group with *H. pylori*-positive gastritis: it was found that patients with both *H. pylori*-related and unrelated gastritis had increased mucosal and total wall thickness compared to the control group, and in particular, those with *H. pylori* gastritis had statistically thicker walls than those with gastritis unrelated to *H. pylori*.

Our study is innovative because we have demonstrated a statistically significantly increased submucosal thickness and submucosa/total thickness ratio in chronic gastritis from *H. pylori*; however, the comparison between mucosal thicknesses was not statistically significant. Furthermore, we were the only ones to assess gastric motility and the presence of luminal ingests; in our study, reduced gastric motility was statistically significant in patients with *H. pylori* gastritis compared to the other two groups; substantial differences in the presence of ingests among the various groups were not observed.

The lack of a post-eradication follow-up to assess the reversibility of gastric wall changes will be explored in a subsequent study. Indeed, this pilot study aims to demonstrate that benign gastric pathologies can result in detectable alterations in the stratification of the gastric wall, as observed through ultrasonographic imaging.

Evidence suggests that *H. pylori* eradication therapy, while pivotal for preventing gastric malignancies, may impact the intestinal microbiota and exacerbate or trigger IBD through mechanisms such as gut dysbiosis and immune modulation.

For patients with IBD undergoing intestinal ultrasound as part of routine evaluation, the addition of a gastric ultrasound could offer a non-invasive means to detect coexistent H. pylori-related gastritis. As such, ultrasonographic findings may contribute to risk stratification and more tailored management strategies in IBD patients, highlighting their potential as a valuable diagnostic and monitoring tool in this population [16].

Several studies have shown that *H. pylori* infection plays a role in delayed gastric emptying due to edema associated with antral gastritis, leading to gastric outlet obstruction, or due to gastroparesis caused by the release of inflammatory and immune mediators [17]. Fock et al. [18] studied 72 patients with dyspepsia and demonstrated a greater delay in gastric emptying in patients with Helicobacter pylori infection compared to those without infection. Chun-Ling Zhang et al. [19] also demonstrated that the gastric emptying half-times of the proximal end, distal end, and entire stomach in *H. pylori*-negative and *H. pylori*-positive groups undergoing eradication therapy were shorter than those in the *H. pylori*-positive group.

The study is characterized by several methodological limitations. The relatively small cohort size of 113 participants limits the statistical power of the analysis, thereby reducing the applicability to larger and more heterogeneous populations. Additionally, the single-center design further restricts the external validity of the results, limiting their generalizability to broader clinical settings. Moreover, ultrasound is an operator-dependent technique, which could result in variability in outcomes based on the operator’s level of expertise, potentially compromising the reproducibility and accuracy of the measurements. Also, a limitation of this study is its inability to accurately assess the extent of the disease, as changes in the gastric body were not evaluated. Finally, reduced gastric motility was assessed, but the methods used (e.g., the presence of gastric content) may not provide quantitative or highly specific data compared to advanced techniques such as scintigraphy, a gold standard.

Indeed, while existing studies have provided valuable insights into the utility of ultrasound in the evaluation of gastritis and gastric motility, further research is warranted to validate these findings and explore additional applications of ultrasound in gastroenterology. Future studies could focus on refining ultrasound techniques and developing standardized protocols for gastric evaluation.

These limitations underscore the necessity for future multicenter studies involving larger patient cohorts and employing standardized imaging protocols across different operators to enhance the reproducibility and robustness of the findings.

By considering these additional points, we can further enhance our understanding of the role of ultrasound in the evaluation of gastritis and its potential impact on clinical practice and patient care.

## 4. Methods

### 4.1. Patients Recruitment

Written informed consent was obtained from all participants. The prospective observational case–control study involved the enrolment of patients who, due to various symptoms (dyspepsia, epigastric pain, familiarity, anemia, etc.) were referred to the Gastroenterology and Digestive Endoscopy Unit of the “Casa Sollievo della Sofferenza” Hospital in San Giovanni Rotondo (FG) from January 2022 to May 2023 and underwent an esophagogastroduodenoscopy (EGDS). The patients were divided into three groups:-Group 0: control group, patients without evidence of gastritis nor *H. pylori* infection on histological examination;-Group 1: patients with positive histology for gastritis in the absence of *H. pylori*;-Group 2: patients with positive histology for *H. pylori* gastritis.

All patients first underwent an ultrasound examination and, on the same day or a few days later, EGDS with biopsy sampling. Inclusion criteria were as follows: dyspeptic patients or with other clinical indication to perform EGDS; age over 18 years old. Exclusion criteria were as follows: history of gastric surgery; known diagnosis of gastric malignancy; known diagnosis of inflammatory bowel disease; previous treatment of *H. pylori* infection; evidence of atrophic gastritis on histology (because a reduced thickness of the mucosal layer would have been a confounding factor in the measurement of the submucosal layer and therefore in the calculation of the submucosal thickness to wall thickness ratio).

We collected clinical and demographic data: age, sex, body mass index (BMI), smoking habits, intake of PPIs, NSAIDs, alcohol, cigarette use, and examen indication.

### 4.2. Ultrasound Procedures

The ultrasound procedure was performed after fasting for at least six hours, before performing the EGDS if scheduled on the same day using a fixed Canon (Otawara, Japan) ultrasound machine with Aplio 300- series (Aplio, San Bruno, CA, USA). The gastric antrum was identified in a longitudinal scan in the epigastrium, using the superior mesenteric vein as a point of reference; the total thickness of the antral wall (mm) from the lumen to the serosa was measured with a linear probe [7]. Subsequently, the thickness of the submucosa (mm) and the mucosa was measured, and the submucosal thickness/parietal thickness ratio was evaluated; gastric motility (present or reduced) and the presence or absence of ingestion into the lumen were also evaluated.

The main features of the bowel wall evaluated by ultrasound include wall thickness, echogenic pattern, vascularity, and motility. Of these, echogenicity and wall thickening are the most important factors for identifying gastrointestinal diseases. The normal bowel wall is stratified into five distinct layers with different echogenicity. Starting from the inside, the first is the interface between the lumen (hyperechoic) and the mucosa (hypoechoic). The following layers are the submucosa (hyperechoic), muscularis (hypoechoic), and the serosa (hyperechoic). Wall thickness is the main ultrasonographic feature of the gut assessed by ultrasound. Measurements of bowel wall thickening should be taken from the external hyperechoic layer (serosa) to the internal hyperechoic layer (interface between mucosa and lumen). In normal conditions, the gastric wall is typically thicker, ranging between 5 and 7 mm (Figure 4).

### 4.3. Endoscopy and Histology

The gold standard for diagnosis of *H. pylori*-related gastritis is represented by EGDS with biopsies. Biopsy sampling was performed according to the Sydney system reference which encompasses two biopsies at the level of the antrum (on the great and small curvature side 3 cm from the pylorus) and of the body (small curvature 4 cm proximal to the notch and one along the great curvature in the middle) and a biopsy at the level of the incisura angularis.

The results of the biopsies allowed the identification of five histological types of report: normal mucosa, chronic active gastritis, chronic inactive gastritis, atrophic gastritis/metaplasia, and other (autoimmune, ischemic); furthermore, the presence or absence of *H. pylori* was reported in the biopsies.

### 4.4. Eradication Regimens

For patients testing positive for *H. pylori*, an eradication regimen was proposed, either concomitant or bismuth quadruple therapy. The decision regarding the type of regimen was leftup to the physician. For bismuth quadruple therapy, patients received three Pylera^TM^ capsules q.i.d.; for concomitant therapy, a treatment regimen consisting of 1000 mg of amoxicillin twice daily (b.i.d.), 500 mg of clarithromycin b.i.d., and 500 mg of metronidazole b.i.d. was implemented, accompanied by 40 mg of pantoprazole b.i.d. as a proton pump inhibitor. The therapy duration for both regimens was 10 days. Eradication of the infection was assessed 6 to 8 weeks post-treatment using a urea breath test, with the side effects systematically documented. Patient adherence to the regimen was defined as the intake of at least 80% of the prescribed medication. Eradication rates were analyzed and reported both on an intention-to-treat and per-protocol basis.

### 4.5. Statistical Analysis

The clinical and demographic variables of the study population were expressed as absolute values and percentages (for categorical variables) or median, interquartile range (IQR), and mean ± standard deviation (for continuous variables).

The normality of distribution of continuous variables was evaluated by the Kolmogorov–Smirnov test, and the non-parametric test was adopted consequently.

For comparison of categorical variables between independent samples, the Pearson Chi-square Test or the Fisher Exact Test were used when necessary. For comparison of continuous variables, the non-parametric Kruskal–Wallis test for independent samples was used.

All tests were two-tailed.

Variables that were statistically significant in the univariate analysis were tested in the multivariate analysis using a logistic regression model with backward stepwise term selection.

In order to best distinguish patients with ultrasound signs of gastritis from patients without ultrasound signs of gastritis, a cut-off value of the submucosal thickness expressed in mm and calculated using a ROC (receiver operating characteristic) curve was identified. The cut-off is the value that maximizes sensitivity and specificity.

All analyses were performed using SPSS v.13 software and *p*-values < 0.05 were considered significant.

## 5. Conclusions

Despite the gold standard for diagnosing *H. pylori* infection being biopsy sampling via esophagogastroduodenoscopy, there are inherent challenges and limitations associated with this invasive procedure. Advancements in non-invasive diagnostic methods such as the urea breath Test (UBT) and fecal antigen testing have significantly improved the diagnostic process, offering high sensitivity and specificity. Such non-invasive methods provide valuable alternatives, particularly in settings where invasive procedures may not be feasible or desired by patients.

Our study has shown that certain ultrasound characteristics can be predictive of *H. pylori*-related gastritis. The results of our work suggest that thickening of the submucosal layer and reduced gastric motility are statistically significantly associated with *H. pylori* gastritis. These findings are clinically relevant as they suggest that ultrasound could be an effective method for the evaluation and early diagnosis of *H. pylori*-related gastritis, enabling timely therapeutic interventions and improving clinical management of patients with this condition and more widely, including patients with dyspepsia.

## Figures and Tables

**Figure 1 antibiotics-14-00012-f001:**
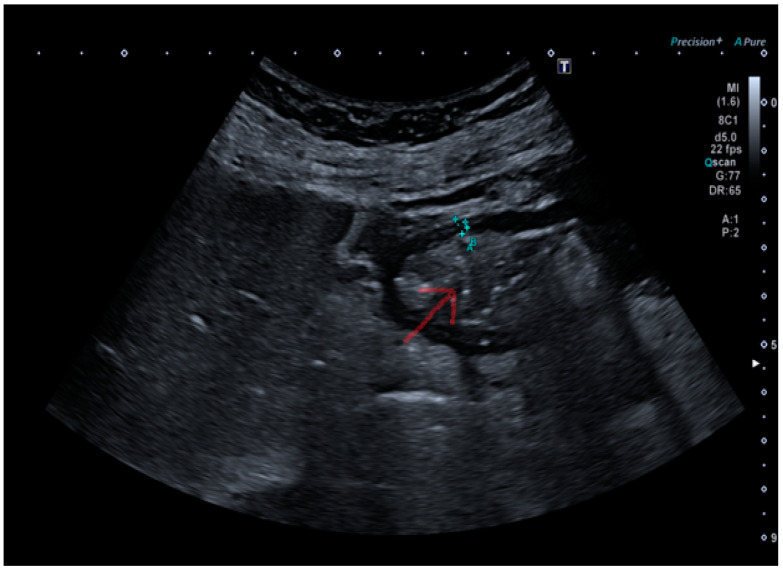
Stomach explored by convex probe containing food ingests in *Helicobacter pylori*-positive patient. The red arrow in the image indicates the presence of ingested material in the stomach.

**Figure 2 antibiotics-14-00012-f002:**
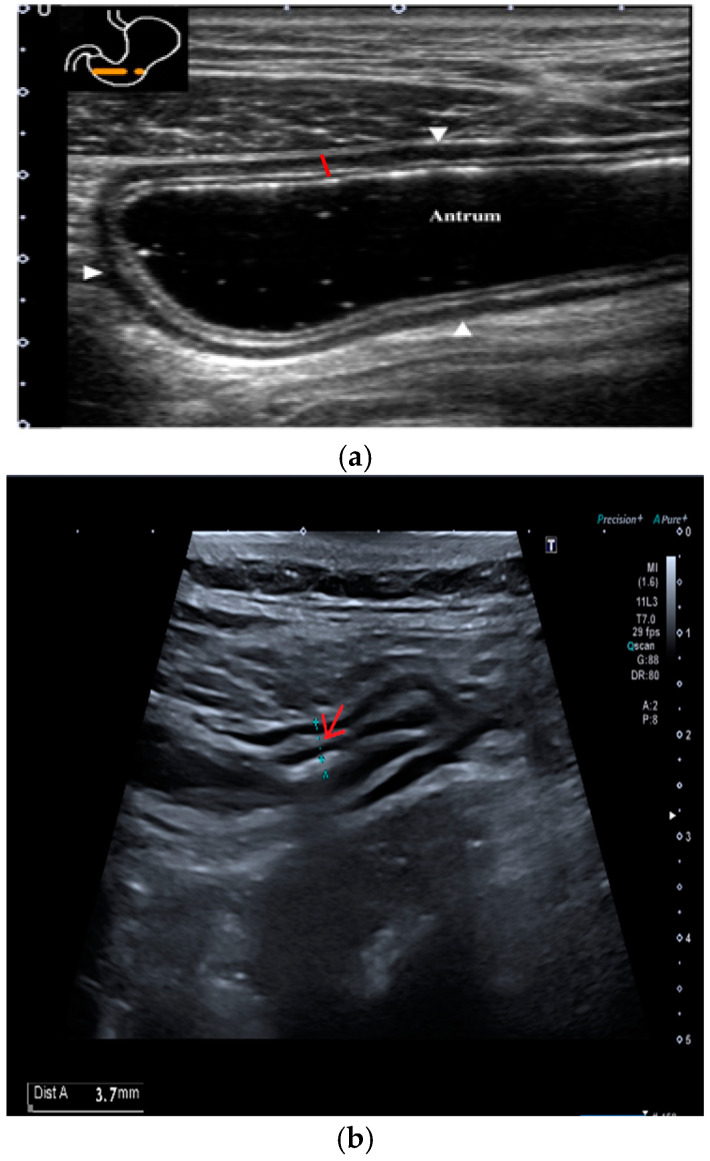
Transversal section of gastric antrum with different layers of the wall. The red line represents the anterior wall of the antrum, showing the normal alternation of layers. Starting from inside: lumen–mucosa interface, mucosa, submucosa, muscularis, and serosa. (**a**) Antrum in a patient without Hp-related gastritis, with a normal submucosal layer. (**b**) Antrum in a patient with Hp-related gastritis, showing a thickened submucosa, marked by a red arrow.

**Figure 3 antibiotics-14-00012-f003:**
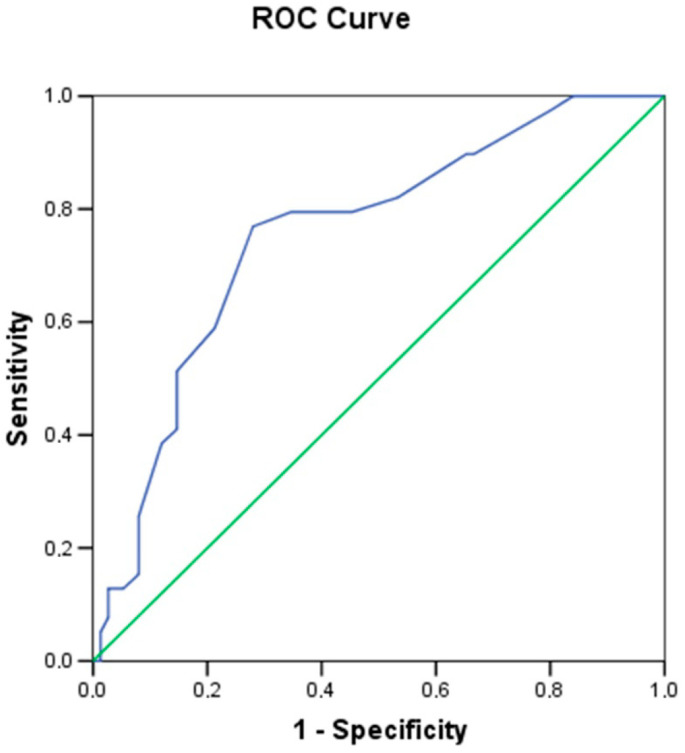
ROC curve between group 1 (green line) and group 2 (blue line).

**Figure 4 antibiotics-14-00012-f004:**
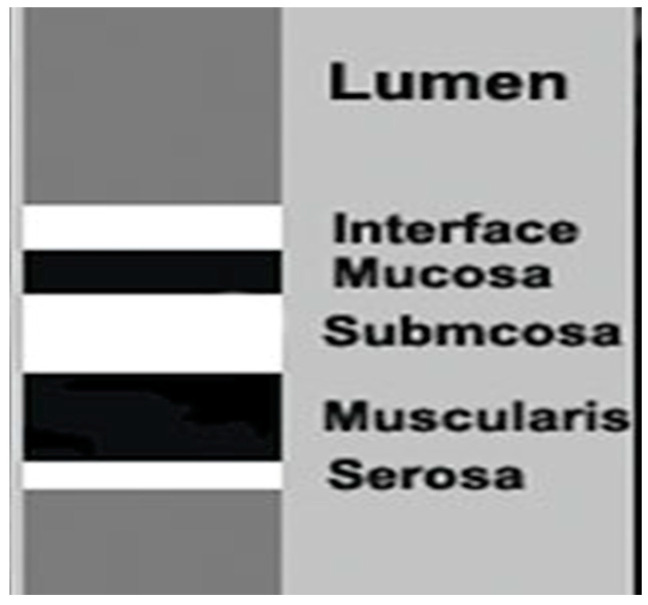
Schematic representation of the alternation of the layers of the gastric wall.

**Table 1 antibiotics-14-00012-t001:** Demographic characteristics.

	Group 0 (=35)	Group 1 (=40)	Group 2 (=38)	*p*
Sex (n, %)				0.857
M	12 (34%)	16 (40%)	15 (39%)	
F	28 (66%)	24 (60%)	23 (61%)	
Age (Mean)	50.00 ± 15.31 (39–62)	54.1 ± 15.09 (45.5–65.5)	57.29 ± 15.58 (50–69)	0.092
Smoking (n, %)				0.189
No	28 (80%)	27 (68%)	32 (84%)	
Yes	7 (20%)	13 (33%)	6 (16%)	
Alcohol (n, %)				0.599
No	25 (71%)	26 (65%)	24 (63%)	
Yes	5 (14%)	11 (28%)	10 (26%)	
Occasional	5 (14%)	3 (8%)	4 (11%)	
PPI (n, %)				0.160
No	16 (46%)	21 (53%)	21 (57%)	
Yes	18 (51%)	18 (45%)	11 (30%)	
Occasional	1 (3%)	1 (3%)	5 (14%)	
NSAID (n, %)				0.367
No	25 (71%)	29 (73%)	20 (53%)	
Yes	8 (23%)	8 (20%)	14 (37%)	
Occasional	2 (6%)	3 (8%)	4 (11%)	

**Table 2 antibiotics-14-00012-t002:** Ultrasound characteristics of the three enrolled groups.

	Group 0 (=35)	Group 1 (=40)	Group 2 (=38)	*p*
SLT (mm)	1.22 ± 0.39 (1.00–1.40)	1.52 ± 0.74 (1.00–1.80)	1.89 ± 0.64 (1.60–2.10)	0.000004
MLT (mm)	1.11 ± 0.40 (0.90–1.30)	1.00 ± 0.38 (0.80–1.80)	1.03 ± 0.52 (0.70–1.20)	0.279
SLT/AWT Ratio	0.28 ± 0.07 (0.25–0.33)	0.34 ± 0.10 (0.26–0.40)	0.40 ± 0.09 (0.33–0.49)	0.000002
Reduced motility (n, %)				
No	19 (54%)	21 (53%)	10 (26%)	
Yes	16 (36%)	19 (48%)	28 (74%)	0.024
Delayed gastric emptying (n, %)				
No	23 (66%)	26 (65%)	18 (74%)	
Yes	12 (34%)	15 (45%)	20 (54%)	0.185

**Table 3 antibiotics-14-00012-t003:** Multivariate analysis.

		*p* Value	OR	95% CI for OR
SLT (mm)	Group 1	0.015	3.364	1.29–10.241
SLT (mm)	Group 2	<0.001	8.592	2.90–25.455

**Table 4 antibiotics-14-00012-t004:** Antibiotic therapy in group 2 patients.

Therapy	Number of Patients	Eradication Rate
BISMUTH QUADRUPLE THERAPY	29	28 (96.5%)
CONCOMITANT THERAPY	6	6 (100%)

## Data Availability

Data are contained within the article.

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
