# Peer review of "Ultrasound Features of Helicobacter pylori-Related Gastritis"

_antibiotics, 2024, doi:10.3390/antibiotics14010012_

Round 1
Reviewer 1 Report
Comments and Suggestions for Authors
This compelling study investigated the ultrasonographic features of gastritis caused by H. pylori infection. To date, gastric ultrasonography is not included in the standard management of this infection, which highlights the novelty of this work.
The authors identified an SLT cut-off of 1.55 mm with a certain sensitivity and specificity to distinguish H. pylori-positive patients. While this finding is not sufficiently conclusive to alter the current approved diagnostic management of this infection, it is certainly of interest.
Here are a few minor revision suggestions:
- Recently, attention has been drawn to the conditions under which H. pylori eradication may be warranted in patients with known inflammatory bowel disease (IBD) (https://pubmed.ncbi.nlm.nih.gov/39195178/). It would be interesting to discuss this study, as the identification of ultrasonographically positive gastritis might stimulate the need for eradication and could also facilitate non-invasive assessment of IBD activity.
- In some instances, H. pylori is not italicised; consistency should be ensured.
- It would be helpful to specify in the statistical analysis section whether a test was conducted to assess the normality of continuous variables, to clarify the decision to report exposure as mean ± SD or as median (IQR).
- Additionally, the statistical analysis section should clarify whether the reported P-values were one-tailed or two-tailed.
- It might be appropriate to expand the discussion on the limitations of this study and the modest results in terms of the specificity and sensitivity of the technique in your setting.
Author Response
Reviewer 1
This compelling study investigated the ultrasonographic features of gastritis caused by H. pylori infection. To date, gastric ultrasonography is not included in the standard management of this infection, which highlights the novelty of this work.
The authors identified an SLT cut-off of 1.55 mm with a certain sensitivity and specificity to distinguish H. pylori-positive patients. While this finding is not sufficiently conclusive to alter the current approved diagnostic management of this infection, it is certainly of interest.
Here are a few minor revision suggestions:
- Recently, attention has been drawn to the conditions under which H. pylori eradication may be warranted in patients with known inflammatory bowel disease (IBD) (https://pubmed.ncbi.nlm.nih.gov/39195178/). It would be interesting to discuss this study, as the identification of ultrasonographically positive gastritis might stimulate the need for eradication and could also facilitate non-invasive assessment of IBD activity.
Answer: Evidence suggests that H. pylori eradication therapy, while pivotal for preventing gastric malignancies, may impact the intestinal microbiota and exacerbate or trigger IBD through mechanisms such as gut dysbiosis and immune modulation.
For patients with IBD undergoing intestinal ultrasound as part of routine evaluation, the addition of gastric ultrasound could offer a non-invasive means to detect coexistent H. pylori-related gastritis. As such, ultrasonographic findings may contribute to risk stratification and more tailored management strategies in IBD patients, highlighting its potential as a valuable diagnostic and monitoring tool in this population
We thank the reviewer for his useful comment. The suggested reference was integrated in the revised text.
- In some instances, H. pylori is not italicised; consistency should be ensured.
Answer: We performed the requested change.
- It would be helpful to specify in the statistical analysis section whether a test was conducted to assess the normality of continuous variables, to clarify the decision to report exposure as mean ± SD or as median (IQR).
Answer: Normality of distribution of continuous variables was evaluated by Kolmogorov-Smirnov test and non-parametric test were adopted consequently.
- “Additionally, the statistical analysis section should clarify whether the reported P-values were one-tailed or two-tailed.”
Answer : ” All tests were two-tailed”
- It might be appropriate to expand the discussion on the limitations of this study and the modest results in terms of the specificity and sensitivity of the technique in your setting.
Answer:
“The study is characterized by several methodological limitations. The relatively small cohort size of 113 participants limits the statistical power of the analysis, thereby reducing the applicability to larger and more heterogeneous populations. Additionally, the single-center design further restricts the external validity of the results, limiting their generalizability to broader clinical settings. Moreover, ultrasound is an operator-dependent technique, which could result in variability in outcomes based on the operator's level of expertise, potentially compromising the reproducibility and accuracy of the measurements. As well, a limitation of this study is the inability to accurately assess the extent of the disease, as changes in the gastric body were not evaluated. Finally, reduced gastric motility was assessed, but the methods used (e.g., the presence of gastric content) may not provide quantitative or highly specific data compared to advanced techniques such as scintigraphy, a gold standard.
Ultrasound is not directly comparable to non-invasive diagnostic methods for H. pylori infection in terms of sensitivity and specificity, as it assesses different pathological aspects. Unlike direct diagnostic tests, which identify the presence of the pathogen, ultrasound evaluates the structural consequences of H. pylori-related gastritis, specifically an increased thickness of the submucosal layer. This technique primarily focuses on structural alterations induced by inflammation, particularly when it is most pronounced in the antral region”
Reviewer 2 Report
Comments and Suggestions for Authors
The article " Ultrasound features of Helicobacter pylori related gastritis" have evaluated the Ultrasound (US) features of H. pylori gastritis and to study the role of US to predictive of the disease. Although it is not the very first study that studied gastritis through this angle, this study carries merits on its own.
This study has a greater number of subjects than ever studied before. It has found that at least thickening of the submucosal layer and a decreased of gastric motility (measured through UT) can be predictive of the disease. But, at the same time the sensitivity and specificity of the methods seem below the current gold standard methods. The study has several limitations that authors have duly accepted but with improvement in quality and sensitivity of US, in near future the course of gastritis can be better understood by non-invasive methods. Kudos to authors for this study as a baseline study for any possible future multi-centered large studies.
Although article is well-written covering minute to minute technical aspects and it seems the article I received is already a reviewed article with changes made been highlighted, still few minor changes for improvement of the article are as follows:
1. Please incorporate later part of conclusion at the end of the discussion, so that conclusion is concise and to the point based on the results.
2. I would suggest to elaborate more limitations of the study which will help readers to understand the importance of this article. Or else, few lines can be incorporated in discussion the highlight key future prospective that can be drawn from this article, how findings of this article can be beneficial for diagnostic prospective of patients. While discussing its always better to compare US with current non-invasive methods in terms of sensitivity and specificity. Also, we can compare the cost effectiveness and feasibility of the US in the diagnosis.
3. Please consider if the article requires ethical clearance although consents have been taken.
4. Please use consistent affiliation for all authors.
Author Response
Reviewer 2
The article " Ultrasound features of Helicobacter pylori related gastritis" have evaluated the Ultrasound (US) features of H. pylori gastritis and to study the role of US to predictive of the disease. Although it is not the very first study that studied gastritis through this angle, this study carries merits on its own.
This study has a greater number of subjects than ever studied before. It has found that at least thickening of the submucosal layer and a decreased of gastric motility (measured through UT) can be predictive of the disease. But, at the same time the sensitivity and specificity of the methods seem below the current gold standard methods. The study has several limitations that authors have duly accepted but with improvement in quality and sensitivity of US, in near future the course of gastritis can be better understood by non-invasive methods. Kudos to authors for this study as a baseline study for any possible future multi-centered large studies.
Although article is well-written covering minute to minute technical aspects and it seems the article I received is already a reviewed article with changes made been highlighted, still few minor changes for improvement of the article are as follows:
- Please incorporate later part of conclusion at the end of the discussion, so that conclusion is concise and to the point based on the results.
Answer à We performed the requested change.
- I would suggest to elaborate more limitations of the study which will help readers to understand the importance of this article. Or else, few lines can be incorporated in discussion the highlight key future prospective that can be drawn from this article, how findings of this article can be beneficial for diagnostic prospective of patients. While discussing its always better to compare US with current non-invasive methods in terms of sensitivity and specificity. Also, we can compare the cost effectiveness and feasibility of the US in the diagnosis.
We thank the reviewer for the useful comment. The following discussion wasa integrated into the revised manuscript. “The study is characterized by several methodological limitations. The relatively small cohort size of 113 participants limits the statistical power of the analysis, thereby reducing the applicability to larger and more heterogeneous populations. Additionally, the single-center design further restricts the external validity of the results, limiting their generalizability to broader clinical settings. Moreover, ultrasound is an operator-dependent technique, which could result in variability in outcomes based on the operator's level of expertise, potentially compromising the reproducibility and accuracy of the measurements. As well, a limitation of this study is the inability to accurately assess the extent of the disease, as changes in the gastric body were not evaluated. Finally, reduced gastric motility was assessed, but the methods used (e.g., the presence of gastric content) may not provide quantitative or highly specific data compared to advanced techniques such as scintigraphy, a gold standard.”
Ultrasound is not directly comparable to non-invasive diagnostic methods for H. pylori infection in terms of sensitivity and specificity, as it assesses different pathological aspects. Unlike direct diagnostic tests, which identify the presence of the pathogen, ultrasound evaluates the structural consequences of H. pylori-related gastritis, specifically an increased thickening of the submucosal layer. This technique primarily focuses on structural alterations induced by inflammation, particularly when it is most pronounced in the antral region”
- Please consider if the article requires ethical clearance although consents have been taken.
Answer à The study was approved after an internal review of the board of the Gastroenterology Unit of “Casa Sollievo della Sofferenza”.
- Please use consistent affiliation for all authors.
Answer à We performed the requested change
Reviewer 3 Report
Comments and Suggestions for Authors
1. The study authors did not include follow-up ultrasound assessments after H. pylori eradication therapy, and therefore, in the results we have now, it is not clear whether ultrasound markers (e.g., submucosal thickening, decreased motility) are reversible after treatment or remain as residual features. Since this is important information for clinicians, I believe the authors should address this issue in their article.
2. Factors such as proton pump inhibitor (PPI) use, smoking and NSAID use, were not controlled for in the statistical analyses in the article. These factors are very common in daily life and may independently affect gastric wall thickness or motility, confounding the results. Because of that, the authors should address these factors in more detail and include them in the analysis.
Author Response
Reviewer n. 3
- The study authors did not include follow-up ultrasound assessments after H. pylori eradication therapy, and therefore, in the results we have now, it is not clear whether ultrasound markers (e.g., submucosal thickening, decreased motility) are reversible after treatment or remain as residual features. Since this is important information for clinicians, I believe the authors should address this issue in their article.
Answer à The reviewer pointed out an interesting point. The lack of post-eradication follow-up to assess the reversibility of gastric wall changes will be explored in a subsequent study. Indeed, this pilot study aims to demonstrate that benign gastric pathologies can result in detectable alterations in the stratification of the gastric wall, as observed through ultrasonographic imaging. Reversibility of such ultrasonographic features after bacterial eradication will be investigated in a future study, which is ongoing in our center.
Factors such as proton pump inhibitor (PPI) use, smoking and NSAID use, were not controlled for in the statistical analyses in the article. These factors are very common in daily life and may independently affect gastric wall thickness or motility, confounding the results. Because of that, the authors should address these factors in more detail and include them in the analysis.
Answer à PPI, smoking and NSAID use were included in the univariate analysis; however, as they did not reach statistical significance. Therefore, they were not considered in the multivariate analysis.
Round 2
Reviewer 1 Report
Comments and Suggestions for Authors
The authors performed the requested changes. No further comments.
Reviewer 2 Report
Comments and Suggestions for Authors
Sincere thanks for revising the article "Ultrasound features of Helicobacter pylori related gastritis". Authors have addressed most of the reviewer’s recommendations.
Reviewer 3 Report
Comments and Suggestions for Authors
I don't have any new comments or suggestions.